# Review on Spinning of Biopolymer Fibers from Starch

**DOI:** 10.3390/polym13071121

**Published:** 2021-04-01

**Authors:** Selamu Temesgen, Mirko Rennert, Tamrat Tesfaye, Michael Nase

**Affiliations:** 1Institute for Biopolymers and Sustainability (ibp), Hof University of Applied Sciences, 95028 Hof/Saale, Germany; mirko.rennert@hof-university.de (M.R.); michael.nase@hof-university.de (M.N.); 2Ethiopian Institute of Textile and Fashion Technology, Bahir Dar University, Bahir Dar P.O. Box 1037, Ethiopia; tamrat_tsfy@yahoo.com; 3School of Textiles, Kombolcha Institute of Technology, Wollo University, Kombolcha P.O. Box 208, Ethiopia

**Keywords:** bio-based materials, biofibers, biopolymers, oil-based polymers, starch, spinning

## Abstract

Increasing interest in bio-based polymers and fibers has led to the development of several alternatives to conventional plastics and fibers made of these materials. Biopolymer fibers can be made from renewable, environmentally friendly resources and can be fully biodegradable. Biogenic resources with a high content of carbohydrates such as starch-containing plants have huge potentials to substitute conventional synthetic plastics in a number of applications. Much literature is available on the production and modification of starch-based fibers and blends of starch with other polymers. Chemistry and structure–property relationships of starch show that it can be used as an attractive source of raw material which can be exploited for conversion into a number of high-value bio-based products. In this review, possible spinning techniques for the development of virgin starch or starch/polymer blend fibers and their products are discussed. Beneficiation of starch for the development of bio-based fibers can result in the sustainable replacement of oil-based high-value materials with cost-effective, environmentally friendly, and abundant products.

## 1. Introduction

The effects of oil-based plastics and additives on climate, water and soil quality are omnipresent. In the last few decades, many scientific studies and developments have been undertaken in order to manufacture products sustainably from renewable raw materials and thus reduce the dependence on fossil raw materials and the global amount of microplastics. Textiles based on synthetic fibers are supposed to be one of the biggest sources for microplastic due to washing abrasion and the contamination of wastewater. The world fiber market accounted for the largest percentage of synthetic fibers by the early 2000s, which has increased rapidly in comparison to 5% in 1960 [1]. Before the usage of synthetic polymers, the major sources of fibers were cotton, wool, and silk. The market share of cotton fiber dropped from 81% to 39% over the last century due to high demand for synthetic fibers [1]. Excessive dependence on oil-based polymers has negative impact, because the sustainability of synthetic materials is limited by natural oil deposits. The production and disposal of oil-based polymers causes environmental pollution. The need to use bio-based materials to substitute synthetics becomes more relevant for the demand to a circular bio-economy.

Biopolymers or bio-based polymers are renewable, environmentally friendly and biodegradable materials, which can be used as an ideal substitute for oil-based polymers that are synthetic in nature [2]. The increasing demand in bio-based polymers/biopolymers has brought an endeavor to develop several alternatives to synthetic materials or synthetic polymers [3]. One of the interesting alternatives to synthetic materials/polymers is the use of natural raw materials such as starch, lignin, keratin, chitosan, gelatin, collagen, and cellulose for biopolymer synthesis. Starch is a naturally abundant biopolymer and has the potential for mass commercial use. Starch has often been used as a food ingredient, although it has become a good alternative for industrial applications in areas such as paper, pharmaceuticals, and textiles [4].

Starch is the predominant carbohydrate reserve in many plants such as potato, tapioca, and cornstarch [5]. Starch has a semicrystalline morphology with different degrees of crystallinity. It consists of two major components: amylose, mostly linear α(1_4) glucan linkage; and amylopectin, an α(1_4) glucan linkage which has α(1_6) linkages at the branch point [6]. The association of hydroxyl groups among the different molecules has brought on different properties of the two forms of starch, form (1) amylose and form (2) amylopectin (see Figure 1). Structurally, amylose is linear with 2–5 relatively long branches and the branches have average polymerization degrees of about 350 monomer units. The average molecular weight of linear amylose molecules is from 0.2–2 million g/mol, while the molecular weight of branched amylopectin molecules is as high as 100–400 million g/mol [7].

The linear nature of amylose can facilitate the orientation and parallel alignments of hydroxyl groups. The occurrence depends on the conditions for macromolecular chain mobility, particularly by dilution with suitable solvents in most cases. Alternatively, dilution under high temperatures can be applied. By aligning neighboring amylose molecules, the intermolecular interactions can be influenced by the formation of hydrogen bonds. The average polymerization degree of the macromolecular branches in amylopectin molecules is about 25 monomer units. Amylopectin molecules have limited chain mobility and do not align and associate spontaneously due to the highly branched structure of these molecules [7]. Figure 1 illustrates the schematic structure of amylose and amylopectin in starch.

Starch without additives is a biodegradable, non-toxic polymer with excellent biocompatibility, but hardly processable without plasticizers due to its complex macromolecular structure and thus limited ability to react to external deformations [8]. Starch is an abundantly available biomass which is capable of substituting synthetic polymers. Starch in its native form has some limitations related to its mechanical properties (modulus, fracture toughness) and thermal stability [5,9]. Researchers have developed methods to minimize the limitations of starches in order to improve their properties for industrial application [9]. Various processes including plasticization, and physical, chemical enzymatic and genetic modification have been studied [10]. As a conclusion, blends of starch and other biodegradable polymers have been identified as the most promising way to minimize the limitations of native starches to develop blends of starch and biodegradable polymers to obtain materials with a wide range of application [11].

Many biopolymers have been used to form fibers using different spinning techniques [5]. Fiber-reinforced polymer composites have been used as lightweight solutions for commercial industries because of their versatility and specific properties. They have been regarded as advantageous over traditional engineering materials, such as steel or aluminum. The demand for composites is expected to increase from GBP 2.5 billion in 2015 to GBP 10 billion in 2030 in the United Kingdom [12]. Contemporary life cycle assessments and bioeconomic strategies consider local bio-based resources, and carbon sequestration at end-of-life of a product by recycling or targeted degradability if needed.

Starch was identified as an appropriate substitute for the polyester fiber production. Advantages of starch-based fibers are the global availability and low cost of starch, its biodegradability, and potential for a sustainable production [13]. Besides the abovementioned advantages, starch has functional advantages compared to synthetic polymers. It has a better hydrophilicity, biocompatibility, bio-absorbability, and full compostability.

Very few reports have been written about starch-based fiber spinning, focusing only on electrospinning. Further literature is available only on patents, describing the applications of different fiber spinning technologies. In this report, we review the possibilities of various spinning techniques that could be used in the production of bio-based fiber from virgin starch and starch/polymer blends and fibers produced thereof. Bio-based fibers can result in the sustainable replacement of oil-based high-value materials with cost-effective, environmentally friendly, and abundant products. Therefore, this paper is expected to be good source of information for those working with the fabrication of fibers from starches and starch/polymer blends.

## 2. Starch as a Source of Fiber

Starch is one of the most important renewable resources and is considered as the second largest source of biomass on earth after cellulose [14]. The major sources of starch are maize, rice, wheat, potato, cassava, yams, and sorghum [15]. Starch-based fibers have been fabricated mainly by blending native or modified starches with polymers, plasticizers, cross-linkers, or other additives [16,17]. Additionally, more recently, native starch has also been spun into fibers using different spinning technologies [17,18,19]. Spinning has a long tradition and is supposed to be the best method for continuous fiber production. The first starch-based fiber application was made using amylose. In this attempt to spin fiber from starch, it was assumed that the linear character of starch macromolecules built side-by-side associations with the amylose monomers under favorable conditions. In 1970, Barger et al. developed a method to produce amylostic filaments and fibers. These researchers developed a method of producing amylostic filaments and fibers having unique physical properties for various technical applications. The method used by these researchers is forming a physical mixture of amylostic solids and controlled amount of water and then subjecting this mixture to higher temperatures and pressures in order to form a jelly-like plastic mass. From their work, it was demonstrated that they successfully extruded the plastic mass into filaments having desirable characteristics of strength, flexibility, and water solubility.

Hiemstra and Muetgeert [18] established a continuous process for the extrusion of adjusted amylose. The extrusion was carried out on a watery sodium hydroxide solution with concentrated ammonium sulfate having a concentration greater than 30 wt.% in the coagulation bath. The ammonium sulfate used in this process was later removed by washing. In another study, a Viscose spinning machine was utilized for spinning amylose fibers from starch [20]. In this study, 5% NaOH watery starch solution was put into a coagulating bath together with sulfuric acid and sodium sulfate at 50 °C. Kunz demonstrated the possibility of spinning amylose fibers, but the claimed caustic amylose composition failed to produce fibers with sufficient tensile strength. In another recent patent by Zussman et al. [19], high-amylose starch-formate was electrospun into fibers. According to these researchers, the method comprises two essential steps: the demonstration of a first spinning dope, which comprises a solution or dispersion of starch in a solvent with at least 50% vol. formic acid and the right amount of starch above the critical entanglement concentration, where starch fibers are supposed to be produced; and secondly, the electrospinning dope to produce a starch-formate fiber.

Spinning amylose suffered from different disadvantages such as insufficient fiber tensile strength and cost-intensive adjusted amylose from starch composition, because starches have a higher content of amylopectin [17]. Studies are available on various methods of fiber spinning from starch compositions containing both amylopectin and amylose components [18,19]. Spinning fibers from starches generally consisted of two categories of starch compositions, namely thermoplastic starch composition (TPSC) and non-thermoplastic starch compositions (NTSC).

### 2.1. Fiber Spinning from Thermoplastic Starch Compositions

Thermoplastics are uncrosslinked polymers and have a low-viscous flow once heated above a specific melting temperature *T_m_* and become hard when they are cooled below *T_m_* with a complete chain immobility if cooled under the glass transition temperature *T_g_*. Thermoplastics can undergo plastic deformation due to their large molar mass, entanglements, intra- and intermolecular interactions, and chain branches. The morphology of thermoplastics can be fully amorphous (only *T_g_*) or semicrystalline with *T_g_* and *T_m_* [21]. Amorphous thermoplastics include mostly translucent plastics, which have a high-entropic molecular structure with a sharp softening point. Amorphous thermoplastics have isotropic rheological properties and a better dimensional stability without shrinkage, good impact resistance, and poorer fatigue resistance compared to semi-crystalline plastics. Amorphous thermoplastics also tend to have lower chemical resistance due to higher free volume and lower tough bonds, such as in semi-crystalline polymers. The characteristics of semi-crystalline thermoplastics is that they have highly ordered molecular structures with sharp melting points, unlike amorphous thermoplastics. Due to the phase boundaries between amorphous and crystalline areas, semi-crystalline polymers appear rather opaque.

Thermoplastic starch can have both an amorphous or semi-crystalline morphology, containing gelatinized or destructured starch in the presence of plasticizers. Fed into an extruder with external heat and applied shear forces of screw–barrel interactions, thermoplastic starch can be molded into plastic products. It can be repeatedly melted and hardened, allowing its processing in common techniques used in the plastic industry. Starches are not truly thermoplastic in nature, which means they have no thermoplastic character in their native form [22]. In the presence of plasticizers such as water, glycerol, sorbitol, etc., at high temperatures above 90 °C and under shear forces, starch can start to melt and flow, which makes it usable for injection molding, extrusion, or blow molding, similar to most conventional synthetic thermoplastic polymers [23]. Figure 2 shows the differences between the morphology of typical starch before and after plasticization with glycerol of different amounts.

Thermoplastic starch (TPS) composition is a translucent amorphous polymer with comparable properties to conventional polymers. It is obtained by compounding native starch with suitable plasticizers at starch gelatinization temperature. Hydrogen bonds present in native starch will be weakened by this operation which leads to fully amorphous material. The resulting material is known as plasticized starch, destructured starch or thermoplastic starch (TPS) [22,23,24]. It can be fully amorphous (TPSA) or semi-crystalline (TPSC).

Attempts to spin thermoplastic starch compositions dates back to 1951. At this time, wet spinning technology was used to extrude the composition of aqueous starch dispersion and glycerol in a methanol/glycerol coagulation bath to obtain starch fibers [10]. The work by Buehler et al., given in U.S. patent Nos. 5516815 and 5316578, describes the application of thermoplastic starch compositions to make starch fibers using melt-spinning processes [26]. In this study, the melted thermoplastic starch was extruded through a spinneret to produce filaments. The filaments were subsequently drawn down mechanically or thermo-mechanically by a drawing unit to reduce the fiber diameter [27]. In another patent by Buehler et al. [10], they produced starch-containing fibers from corn starch and potato starch using different plasticizers. The starch melt-spinnable composition, which was subsequently granulated, had different proportions of starches with varying degrees of substitutions and amylose content. The fiber produced had an average tenacity of around 1.1 cN/tex.

Usually, it is difficult to obtain sufficient properties from semi-crystalline TPSC alone for fiber spinning processes or fiber applications because of weak mechanical properties, retrogradation, and the high hygroscopic nature of thermoplastic starches TPS. Thus, fibers of TPS blends were investigated in much of the literature and patents on spinning TPSC fibers. A lot of research on the application of starch focused on blending thermoplastic starch and other synthetic polymers [16,27,28,29,30]. According to the study by Lorcks et al. [31], mixtures of thermoplastic starch from native potato starch and poly(lactic acid) (PLA), polyesteramide, and a copolyester of aliphatic diols and aliphatic/aromatic dicarboxylic acids was successfully spun into fibers. Bond et al. [29] described the preparation of high elongation multicomponent fibers from starch and other polymers in their patent. In this work, it was revealed that multi-component fibers were configured as sheath–cores, wherein the thermoplastic polymer component constituted the sheath and the thermoplastic starch component constituted the core component.

Biodegradable fibers from starch composites have been developed by different researchers. Nakajima and Taniguchi used polybutylene succinate (PBS) as the core material and a starch/polymer composite as the sheath, and successfully spun fibers [32]. Different kinds of polymers have been used by different researchers mixed with starch for fibers spinning, and the success of spinning these starch/polymer composites into core/sheath fibers was observed [28,29]. Some of the polymers used include Polycaprolactone (PCL), polylactic acid (PLA), polybutylene succinate (PBS), polyvinyl alcohol (PVA) and ethylene vinyl acetate (EVA) [18].

Most of the early attempts made in fiber spinning from thermoplastic starch composition are in the form of patents. Some of thermoplastically processable starch compositions are disclosed in U.S. patent Nos. 4900361, 5095054, 5736586 and PCT publication WO 98/40434 filed by Hanna et al. These abovementioned starch compositions do not contain the high molecular weight polymers which are essential to achieve the required melt viscosity and melt extensibility. The melt viscosity and melt extensibility are essential material characteristics used for production of fine fibers, thin films, or thin-walled foams to avoid fracture and failures [33].

A TPSC composition consisting of dextrin, a kind of destructured starch from tapioca or corn starch, and polyacrylamide (PAM) was developed by Bailey et al. [20]. These researchers used different plasticizers and used melt spinning technology for fiber spinning. These researchers claimed up to 10 wt.% of PAM in the starch recipe, but it was evidenced from their work that even less than 0.1 wt.% seemed to be working well. This TPSC composition can be extruded into fibers and films. The summaries of TPSC fiber spinning patents are illustrated in Table 1.

### 2.2. Spinning Non-Thermoplastic Starch Compositions

Non-thermoplastic starch composition (NTPSC) consists of starch and water softening to a degree that the material can be brought into a flow. This combination can be processed, e.g., by spinning, to form a plurality of non-thermoplastic starch fiber suitable for forming a flexible fibrous structure [34]. Non-thermoplastic starch composition is not processable by melt spinning, but it can be processed using wet spinning and dry spinning. There is risk of decomposition in the case of processing non thermoplastic starch using melt spinning because its degradation temperature is lower than its melting temperature. The non-thermoplastic starch composition also differs from a thermoplastic composition. The non-thermoplastic starch composition is dewatered by drying and obtains a solid state by losing its thermoplastic properties [35]. The non-thermoplastic starch fiber has no melting point because it decomposes before reaching melting temperature.

In early attempts to produce starch fibers, a wet spinning process was used principally. For example, a starch/solvent colloidal suspension can be extruded from a spinneret into a coagulating bath. Previous studies for wet spinning starch fibers include U.S. Pat. Nos. 4139699, 4853168 and U.S. Pat. No. 4234480 [35]. While spinning NTSC, some additives such as crosslinking agents can be included, in order to improve the properties of starch fibers.

Previous work conducted by Eden and Trksak [33] described the possibility of spinning high amylose starch. Some other researchers also used ammonium salt in the coagulation bath to crosslink the final starch and starch/PVA fibers. A process for the dry spinning of non-thermoplastic starch composition is described in recent patents [30]. A new spinning device was invented by Bastioli et al. [36], and in this device starch fibers are formed and coagulated as watery starch dispersions travel through the holes of tubular a wall into a coagulating chamber. The coagulation agent used was ammonium sulfate. The fiber diameter was dependent on the size of the holes, which may have narrow outlets opening 10 to 500 µm.

In a patent claimed by James et al. [35], an NTPSC of unsubstituted polysaccharide and PVA were spun into fibers with an average fiber diameter less than 50 µm. Starch acetate, a modified starch, was also used for production of fibers [24]. It was produced by grafting starch with acetic acid or acetic anhydride, wherein catalysts such as perchloric acid or sulfuric acid were used. Fibers spun from a non-thermoplastic starch composition (NTPSC) are summarized as given in Table 2.

## 3. Possible Spinning Techniques for Starch Biopolymer

### 3.1. Conventional Fiber Spinning Methods

The conventional fiber spinning methods are melt spinning, wet spinning, and dry spinning. These methods have been widely used in synthetic fiber manufacturing. Melt spinning is used for thermoplastic polymers that can be melted. In melt spinning, the polymer melt can be extruded through a spinneret containing a number of holes. The elongational laminar melt flow causes molecular orientation in the machine (drawing) direction that can be even enhanced by drawing and crystallization upon cooling. The schematic drawing for melt spinning is shown in Figure 3.

In dry spinning, a solvent is used to dissolve the raw material to form a solution and the fiber is formed from the polymer solution. In this process, the polymer can be dissolved in a volatile solvent, the solution is pumped through the spinneret to form filaments, and air is used to dry the filaments. Usually, the used solvents in dry spinning are expected to have low boiling point and low latent heat or appropriate volatility, respectively. The schematic for dry spinning is given in Figure 4a. Wet spinning is the oldest spinning process. In wet spinning, a polymer solution with the desired viscosity will be prepared by dissolving the polymer in a solvent. The formed solution is forced through the submerged spinneret into a coagulation bath containing a non-solvent for the polymer. The solvent in the spinning dope is extracted into the coagulation bath, and the filaments solidify. The schematic for wet spinning is given in Figure 4b.

These conventional methods have also been widely used in fiber spinning from virgin starch or starch and other polymer blends [28]. Bond et al. [30] described utilization of the melt spinning technique for fiber spinning from starch and polymers blends. In their work, they produced highly attenuated continuous and stable fibers. Another study by the same researches in 2003 revealed the possibility of producing biodegradable fibers from starch and other biodegradable polymers using a melt spinning system [35].

Other researchers have used the wet spinning method for fiber spinning. Nevertheless, the wet-spun fibers are characterized by a coarse diameter, typically greater than 50 µm, and the large amount of solvent used in this process needs an additional drying step and recovery [36]. Some of the previous studies for wet-spinning starch fibers include U.S. Pat. Nos. 4139699, 4853168 and 4234480 [18]. Some other researchers modified the conventional fiber spinning methods. James et al. [36] developed an air-drawing device to attenuate a non-thermoplastic starch composition extrudate and evaporate the water solvent.

### 3.2. Modern Techniques Used for Starch Fiber Spinning

Besides the conventional spinning methods, different kinds of new spinning techniques have been developed and used for fiber spinning from starch and starch/polymer blends. Some of the novel developments in starch fiber production include electrospinning [19,37,38,39,40], electro wet-spinning [41,42], modified electro spinning [43,44] rotary or centrifugal spinning [45,46,47] and solution blowing [14]. From a feasibility point of view, conventional fiber production techniques are still more preferred, because they allow higher outputs and economic benefits [14]. The next section reviews the modern spinning practices which are available for spinning a starch biopolymer and its blend.

#### 3.2.1. Electrospinning of Fibers from Starch

Electrospinning is a fiber spinning technique that uses a high voltage electrostatic field for the production of fibers with diameters in the range of micrometer to nanometer scale. It has become a widely used fiber formation technique in the last decade [14,38]. The conventional process of fiber spinning is based on the principle of pressure-driven extrusions of a viscous polymer into fibers of diameters ranging from 10 to 500 µm. Electrospinning as a practical technique for producing nanofibers began in 1934, when Formhals patented his first invention related to the process and apparatus for producing artificial filaments using electric charges [48].

The differences between electrospinning and the conventional methods (wet/dry spinning) are in the fundamentally different processes, where electrostatic attractions enable the process and also the aerodynamic drag at the end of the process [14]. The linearity of amylose and its ability to align and aggregate made the fabrication of amylose fibers preferable in early attempts to fabricate starch fibers [18]. Some researchers patented a process for the fabrication of amylose fibers [43]. Amylose can be preferentially oriented into parallel alignments leading to the formation of hydrogen bonds because of its linear nature, which gives it sufficient freedom of macromolecular chain mobility. In contrast, the branched nature of amylopectin cannot produce this free movement. It cannot align and associate readily. Theoretically, it is known that the amylopectin component of starches can affect fiber spinning and the strength of the fibers spun [13,18].

Approaches for the electrospinning of nanofibers from natural starches will be economically feasible on a commercial scale even if starches with a high amylopectin content are utilized, because amylopectin is the major component in any natural starches. On the other hand, amylopectin as the major constituent of the starch and a highly branched polymer affects the fiber’s formation during the electrospinning [40]. Therefore, modification on the electrospinning setup was attempted by many researchers to spin amylopectin-rich starches. This was achieved by Kong et al. by modifying the conventional electro spinning setup [17].

A patent designated by Jennifer et al. [38] described the utilization of an electrospinning technique to produce starch filaments. They described the possibility of producing starch filaments from a mixture of starch, water, plasticizers, and other optional additives such as stabilizers or processing aids. Lancuški et al. [49] produced electrospun fibers from corn starch with a high amylose content using formic acid in different concentrations as a solvent, and subjecting each solution to different aging times, i.e., 2, 6, 24, and 72 h before electrospinning. The characterization of the fabricated fiber mats signifies that dispersions with formic acid concentrations of 100, 90, and 80% resulted in fibers with diameters in the range of 80–300 nm.

According to Fonseca et al. [50], soluble potato starch with normal amylose content has been converted into ultrafine fibers by electrospinning like a neat polymer in a fiber-forming solution. These researchers prepared fiber-forming polymer solutions with 40% soluble potato starch (with amylose content of 32.54 ± 3.65%) and formic acid (75%) as the solvent. The solutions were allowed to age for 0, 24, 48, and 72 h before electrospinning. The result revealed that ultrafine fibers were successfully electrospun from the solution.

However, in another study, glutinous rice starch and tapioca starch were used for fiber spinning using an electrospinning technique. In this research, water was used as the only solvent and polymeric dispersions were stirred at 80 °C for 15 min before electrospinning. From the result, it was revealed that unlike tapioca starch, the glutinous rice starch failed to generate nanofibers during the electrospinning [42,51].

The fabrication of high purity natural tapioca starch (NTS) fibers was reported by Sutjarittangtham et al. [43] using a modified electrospinning technique with a dehydration process by using a −20 °C ethanol collector bath to complement the conventional electrospinning technique. Electrospun fibers with diameters of 1.3–14.5 μm were generated from a simple solution of starch in deionized water with starting concentrations of 3.0 to 5.0 wt.%.

More recently, research conducted by Fonseca et al. [27] revealed the possibility of producing electrospun fibers from native and anionic corn starch with different amylose contents. The electrospun fibers of native and anionic corn starches with regular amylose and high amylose contents were prepared by the electrospinning of starch solutions dissolved in 75% formic acid solvent. The result revealed that fibers produced from modified anionic starches have homogeneous morphologies, whereas fibers from regular corn starches contained droplets and have heterogeneous morphologies, with diameters varying from 70 to 264 nm. Fibers produced from starches having amylose content less than 70% have smooth continuous surfaces.

The schematic electrospinning process is given in Figure 5. Studies performed by different researchers on electrospun starch fibers are summarized in Table 3.

#### 3.2.2. Electrospun Fibers from Starch/Polymer Blends

In order to overcome problems associated with natural starches, starches can be blended with other polymers, chemically crosslinked, or plasticized to improve their properties [18,48]. In this case, the addition of a second polymer is intended to promote entanglement. Likewise, the addition of starch can provide the ability to adjust the surface properties of polymer fibers [50]. The common limitations of native starch are related to their poor mechanical property, thermal stability, and their high hygroscopicity. To overcome the limitations and improve their properties, starches are usually blended with other polymers [51]. Starch blending with other polymers is generally intended for the reduction in production cost and improving the properties of starch [49].

Šukytė et al. [52] prepared nanofibers from potato starch, PVA, and small amounts of ethanol. The results showed that with an amount of 5 wt.% potato starch as a blend partner it was impossible to form nanofibers by the electrospinning process. In this study, it was revealed that even a small amount of ethanol had a significant positive influence on the electrospinning process, although it negatively influenced the nanofibers and the associated web structure. Wang et al. [53] prepared fibers of oxidized starch with PVA of 380 nm by means of electrospinning, demonstrating that a high concentration of starch in solution notably affects the homogeneity of fibers. According to Jukola et al. [54], there is a possibility of producing electrospun nanofibers using PCL and starch. In this study, a high concentration of starch was used, and the result revealed the formation of highly porous scaffolds. Electrospinning with starch in multicomponent solution was realized by Sunthornvarabhas et al. [55] using a solution of polylactic acid (PLA) with dichloromethane (DCM) and a solution with cassava starch dissolved in dimethyl sulfoxide (DMSO). The result revealed a fiber structure with homogenous diameter.

Bicomponent nanofibers from PVA/cationic starch solution were produced by using electrospinning [56]. In this research, ethanol was used as a solvent. The result revealed the successful formation of nanofibers with different thicknesses, with varying properties dependent on the cationic starch. Some other researchers produced fibers from blends of modified starch and other polymers using electrospinning [57]. Starch nanocellulose composite fibers with increased mechanical strength have been prepared by Wang et al. [58] by using nanocellulose as a reinforcing filler and cationic starch as a binding agent (matrix). Other researchers produced a core–sheath compound with fibers from starch-formate solution using coaxial spinning [59]. The produced fibers had mean diameters of 4.13 ± 1.05 µm. Fibers electrospun from blends of starches and other polymers are summarized in Table 4.

#### 3.2.3. Electro Wet Spinning

Kong and Zeigler [17] developed a pure starch microfiber web by modifying the conventional electrospinning technique with a coagulation bath (electro wet spinning). The concept of electro wet spinning is to electrospin pure starch fibers without additional polymers as binders or a matrix. To produce pure starch microfibers, the spinning dope solution was prepared by heating Gelose 80 (amylose content—80%) in 95% dimethyl sulfoxide (DMSO) until the gelatinization of starch started. In this process, ethanol was used as a solvent for the coagulation bath because of its miscibility with DMSO and its non-miscibility with starch. The reason for modification of the electrospinning technique with a coagulation bath was to obtain solid fibers with the evaporation of DMSO, which a critical step due to its non-volatility. The addition of pure ethanol in a coagulation bath resulted in the formation of an amorphous electrospun web and the starch web crystallinity was increased to 43% as the water proportion in the ethanol coagulation bath was increased. This study revealed that the average diameter of electrospun starch fibers was 2.60 µm. The developed electrospun web was heat-treated to improve the crystallinity (size and degrees of crystals). The stability of the braid against water was improved by crosslinking with glutaraldehyde [52]. 

Cardenas et al. [37] also produced microfibers from potato starch using the electro wet spinning technique. These researchers changed different parameters during their study for each treatment and produced starch microfibers with average diameters of 15, 17, 23 and 25 µm. These fibers had a degradation temperature of 304 °C, which indicated stable thermal properties even with very thin fiber diameters. Another comparative study by Cardenas et al. [41] revealed the production of wet electrospun fibers with diameters of less than 50 µm from local and commercial potato starches using ethanol as a solvent. They also studied the spinnability of cationized and uncationized fibers. They concluded that cationized fibers are less heat resistant and non-cationized fibers have a high amorphous morphology.

#### 3.2.4. Centrifugal Spinning

Centrifugal spinning is also known as rotary jet spinning, rotor spinning, and force spinning [61]. Although centrifugal spinning has been widely used in the glass fiber industry for making micrometer-scale glass fibers, the use of centrifugal spinning for producing polymer fibers, especially polymer nanofibers, is relatively new and is an emerging technique [45,46,61]. More recently, FibeRio Technology Corporation commercialized large-scale centrifugal spinning machines (Cyclone FE 1.1 M/S and Cyclone FS 1.1) for the mass production of polymer nanofibers. In centrifugal spinning, the centrifugal force is utilized to drive polymeric jets out of spinnerets [47]. Compared with electrospinning, it combines the advantages of a high production rate, applicability to a broad range of materials, and insensitivity to dielectric constant of materials. Another advantage is that porous and well-aligned structures are produced. Both centrifugal spinning and electrospinning are effective techniques for the production of micro- and nanofibers. Pure starch-based fibers were successfully fabricated by “electro-wet-spinning’’ by Kong and Ziegler [17,18]. The techniques used for a successful pure starch-based electro-wet-spinning were demonstrated to be suitable for starches with amylopectin content below 65% and sensitive to the amylopectin content of starches, and limited the spinnability of amylopectin starches by this process [17].

However, other researchers, Li et al. [61] attempted to produce microfibers by centrifugal spinning technique from various starches such as amylose-rich starch, amylopectin-rich starch, potato starch, and waxy starch. In this study, it was revealed that amylopectin-rich starch-based fibers with an average diameter of sub-microns could be successfully spun from amylopectin-rich native corn and potato starches by centrifugal spinning [46]. The amount of amylopectin of the used native corn and potato starches were determined to be about 68.89 and 73.35%. The produced fibers from amylopectin rich corn starch had a smooth surface, and the fibers obtained from amylopectin-rich potato starch had a rough surface. A method for controlling the surface morphology and topography of centrifugally spun starch-based fibers by adjusting the ratio of amylopectin/amylose in starches and the combination with a hot blast temperature was developed by Li et al. [45]. The effects of hot blast temperature, amylopectin, and amylose on fiber surface morphologies were investigated by using potato and corn starches. The structural analysis of the prepared fibers demonstrated that both corn- and potato-based fibers were amorphous, and some physicochemical changes such as the swelling of granules, loss of order in both amorphous and crystal domains, exudation of amylose, chain rearrangement, and chemical gelatinization occurred during centrifugal spinning. Figure 6 represents the typical schematic process for the representation of centrifugal spinning.

#### 3.2.5. Solution Blow Spinning

Solution blow spinning (SBS) has been used industrially since the end of the 19th century, and it is one of the oldest methods of synthetic fiber production. Solution blow spinning is obtained by the combination of elements of both electrospinning and traditional melt blowing technologies [62]. Compared with melt blowing technologies, solution blow spinning has a broad spectrum of possible raw materials and availability. Compared to electrospinning, SBS has a high yield, a short preparation time, and a high usage value. Solution blowing was developed as an alternative method for making non-woven webs of micro- and nanofibers with diameters comparable to those made by the electrospinning process with the advantage of a high production rate [3].

This process combines wet spinning and dry spinning. In both methods, the polymer solution is extruded through the spinnerets into fibers. Solvent used in this process is removed and drawing of the fibers will be performed to reduce the fiber diameter with a consequent improvement of the mechanical strength. In the case of dry spun solution blowing, the polymer solution is pushed through a spinneret into a heated column, called the spinning tower. Within the tower, the polymer solidifies through evaporation of the solvent. In the case of wet spun solution blowing, the spinneret is placed in a chemical bath. Inside the bath, the polymer is precipitated by dilution or a chemical reaction to form fibers [63].

Solution blowing is quite an innovative technique and is less researched for the production of biopolymer fibers. In solution blowing, a polymer solution is extruded and non-polymer melts. Solution blowing is associated to melt blowing. The only difference lies in the solvent evaporation in the former, rather than melt cooling jet solidification in the latter. Solution blowing is basically an isothermal method. Unlike melt blowing [64], which produces microfibers, solution blowing results in nanofibers [3]. The biggest advantage of solution blowing, apart from its ability to blend biopolymers, is its scalability. There are several parameters where solution blowing depends on the nozzle dimensions, air pressure, collecting distance and viscoelasticity of the polymer solution [3]. Figure 7 shows the schematics process for solution blowing.

### 3.3. Comparison of the Different Fiber Spinning Methods

All spinning techniques above have advantages and disadvantages and should be chosen depending on the application and the associated property requirements. Table 5 summarizes the advantages and disadvantages of each spinning technique for the production of starch fibers.

## 4. Application of Bio-Based Materials

Synthetic polymers are mainly obtained from fossil raw materials (crude oil and fracking gas) as by-products from cracking processes (e.g., naphtha). Fossil resources were created from plant biomass over millions of years. They are therefore not a renewable, but finite resource and need to be substitute by renewable (10^−1^–1 years) bio-based resources. For the production of bio-based polymers, different renewable biogenic raw materials such as sugars and polysaccharides (especially cellulose), proteins and vegetable oils, as well as biogenic by-products from forest and food industries such as lignin and pine resin derivatives have been used so far [65]. By depolymerization to their monomers by bio-refineries with subsequent repolymerization, biopolymers with the identical constitution and properties of synthetic polymers, i.e., drop-in biopolymers such as bio-polyethylene (bio-PE), bio-polyamide (bio-PA), etc., can be created. These polymers can be a one-by-one replacement of fossil-based polymer products, but still cannot solve problems such as the creation of non-biodegradable macro- and microplastics. Alternatively, completely new biopolymers with unique structure–properties relationships have been found and created. One of the most promising alternatives to substitute polyethylene terephthalate (PET) as a transparent plastic bottle packaging application is polyethylene furanoate (PEF), generated from cellulose [68]. These two types of biopolymers are non-biodegradable. Biodegradable plastics made from fossil resources are also considered as biopolymers, e.g., polybutylene adipate terephthalate (PBAT). Sustainability, i.e., using local bio-based resources or carbon capture and biodegradability, have become the most important factors, helping to identify biopolymers from fossil fuel-based polymers. Biodegradable biopolymers can be synthesized from renewable resources, can be used to replace fossil fuel-based-polymers, and are highly demanded where plastic pollution occurs [69,70,71,72]. When compared to fossil fuel-based compounds, the use of natural starting materials for the preparation of bio-based products may result in materials with similar or even improved properties [73].

Bio-based and biodegradable polymers have an extensive range of applications such as pharmaceutical, biomedical, horticulture, agriculture, consumer electronics, automotive, textiles, and packaging. The last application is still the most commonly used application in the polymeric industry, with the highest demand for sustainable solutions [5].

The properties and commercial importance of biopolymers can be improved by blending with (i) other biopolymers, (ii) organic or inorganic functional stabilizers, (iii) fillers or (iv) fibers. The resulting biopolymer blends are called biopolymer composites. These composites have a diverse range of applications in the medicine, electronics, construction, packaging, and automotive sectors [74]. The incorporation of bio-based reinforcement materials into biopolymers greatly improves the mechanical properties such as tensile and impact strengths of resulting composites or stabilizes the composite against premature aging [75,76,77]. These biodegradable polymer composites receive enhanced applications due to their excellent mechanical properties and very good compatibility as well as biodegradability. The most widely used biopolymers for the current development of biopolymer composites are poly(lactic acid) (PLA), cellulose esters, polyhydroxyalkanoates (PHAs), and starch-based plastics [68], as well as polyurethanes [70]. It can be shown that biogenic by-products as additives for bio-composites can improve the properties of the biopolymers. For example, lignin can improve flame retardancy and UV stability [70].

Many bio-based materials have been utilized in the packaging industry. Among the total plastics usage, “packaging” occupies the top position with 41% of all plastic applications [69,78]. Plastic is most often the material of choice for the packaging of food, cosmetics, or pharmaceutical products because of its low cost, light weight, and the excellent protection provided to the packaged product. Even biopolymers are still a niche, although the demand for bio-based materials for packaging is expected to grow to 9.45 million tons by 2023 [71]. Some of the biopolymers used in packaging industry are poly(lactic acid) (PLA) [72], cellulose acetate and cellulose derivates [79], polyhydroxyalkanoate (PHA) [80], proteins [81,82], polybutylene succinate (PBS) [83,84], lipids, and waxes [85] and starch [86,87] Besides packaging application, bioplastics and bio-composites can be used in building and construction materials, and the biomedical, automotive, and textile industries [70]. Along with packaging, plastic textile applications have become more interesting.

### Application of Starch Biopolymers

Among the natural bio-based polymeric materials, starch has been identified as one of least expensive polysaccharides, which is easily processable with conventional plastic processing equipment. Native starch resources are available all over the world and has a huge potential for solid plastics and other functional polymer applications [71,88]. It is considered as one of the most promising natural polymers because of its inherent biodegradability, overwhelming abundance, and annual renewability [5]. Therefore, the development and applications of biodegradable starch-based materials and composites has become more demanding [89].

Starch is commonly used in food and agriculture industries. It can also be used for other applications such as in adhesives and paper binders, textiles, chemical production, or as a feedstock for fermentation [86]. Starch-based biopolymer composites and fibers have been investigated in different areas of applications [53,89,90,91,92,93,94]. The interest to use starch biopolymers in advanced materials applications is accrued from its low cost and abundance. Starch as a macromolecule is also appealing because of its physical, chemical and functional properties such as ease of water dissolution, water retention properties, gelatinization, pasting behavior when subjected to elevated temperatures, and ease of modification to optimize functional properties [76].

Some of the advanced application areas of starches are: porous foam structures [95,96,97,98], wastewater treatment [99,100], filtration [101], tissue engineering [102,103], drug delivery [104,105,106,107,108,109,110], pharmaceutical industry [111,112], antimicrobial films and coatings [113,114,115,116], self-healing polymeric materials [117], electronics [118,119] photonics [120], superhydrophobic surfaces [121], sensors [122,123,124,125] and as an antioxidant additive [126].

## 5. Conclusions and Future Prospects

The properties of natural polymers such as desirable biocompatibility, biodegradability, chemical stability, non-toxicity, balanced mechanical properties, and high porosity have provided unique advantages compared to synthetic polymers. Starch biopolymer, being the second most abundant biopolymer on earth, should be regarded as a valuable resource for fiber manufacturing. Many research attempts have been made to fabricate fibers from starch and starch/polymer blends. Moreover, the inefficient mechanical strength, hydrophilicity, decreased thermal stability, and difficulty in processability still limit the application of starches for industrial uses, and particularly their application for fiber fabrication. To overcome these limitations, various processes including plasticization and physical, chemical, enzymatic, and genetic modifications have been studied. Blending starch with other biopolymers is outlined as a viable alternative to overcome these limitations and can particularly enhance the mechanical and thermal properties of native starch. Besides the blending and modification of starches, new techniques for spinning starch to micro- and nanofibers have also been developed for their application in different fields.

Spinning starch has been performed with additives such as processing aids and without additives. The spinning of native starch has been proved successful for the first time by using electro-spinning. Extensive works are being conducted to fabricate starch fibers from native, modified, and thermoplastic starches using different spinning methods. A number of reviews reported on starch fiber spinning, although most of them cover only a few spinning methods comparatively. This review discussed the recent progress in spinning starch fibers and summarized technologies used in fiber spinning along with their comparison. The review also discussed the progress in applications of starch biopolymers and fibers in areas of different technological applications.

Even though there are attempts made in the fabrication of starch fibers from native starches, extensive research and development works are required to develop appropriate and more efficient spinning technologies for starch biopolymers in order to fully utilize them as a source for a bio-based high-value product. Melt spinning has the potential for continuous fiber extrusion with high outputs. In order to minimize shear fracture and the associated degradation of the starch and breaks of the fibers, future work should focus on starch compounds with alternative, eco-friendly plasticizers. Natural plasticizers are supposed to influence the physical structure of starch-based polymers and the rheological properties. The investigation of several processing parameters such as the drawing ratio and cooling conditions might have a decisive influence on the microstructure, and thus the final fiber properties. The characterization of the processing–structure–property relationships of starch-based fibers is part of the future investigations, and will contribute to life cycle assessments and global bioeconomic strategies.

## Figures and Tables

**Figure 1 polymers-13-01121-f001:**
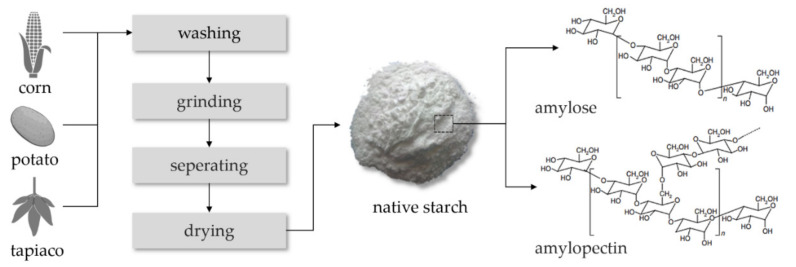
The process of destructured native starch from bio-based resources and the structure of starch with amylose and amylopectin.

**Figure 2 polymers-13-01121-f002:**
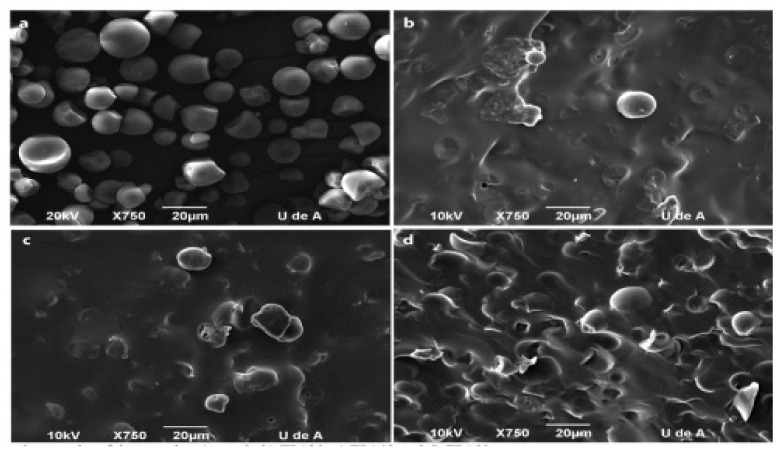
SEM micrographs of the starch and thermoplastic starch (TPS): starch (**a**), TPS30 (**b**) TPS40 (**c**) and TPS50 (**d**). Reprinted with permission from ref. [25]. Copyright 2018 Universidad Nacional de Colombia.

**Figure 3 polymers-13-01121-f003:**
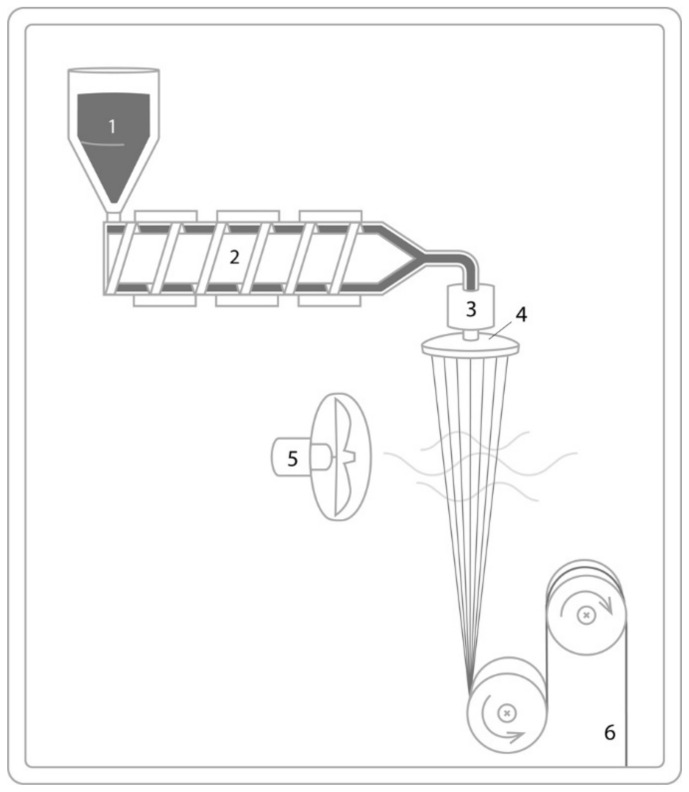
Schematic drawing of melt spinning with a polymer feeder (1), extruder with a single screw (2), the feed pump and nozzle (3), the spinneret (4), cooling air (5) and the winding station (6).

**Figure 4 polymers-13-01121-f004:**
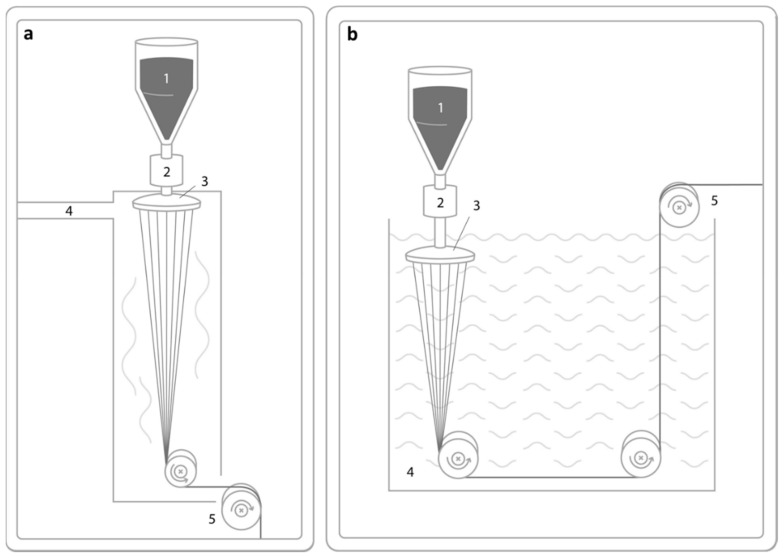
Schematic drawing of dry spinning (**a**) with the polymer solution (1), the pump (2), spinneret (3,) heated chamber (4), and subsequent winding or further processing (5), and the wet spinning process (**b**) with the polymer solution (1), pump (2), spinneret (3), coagulation bath (4) and subsequent winding or further processing (5).

**Figure 5 polymers-13-01121-f005:**
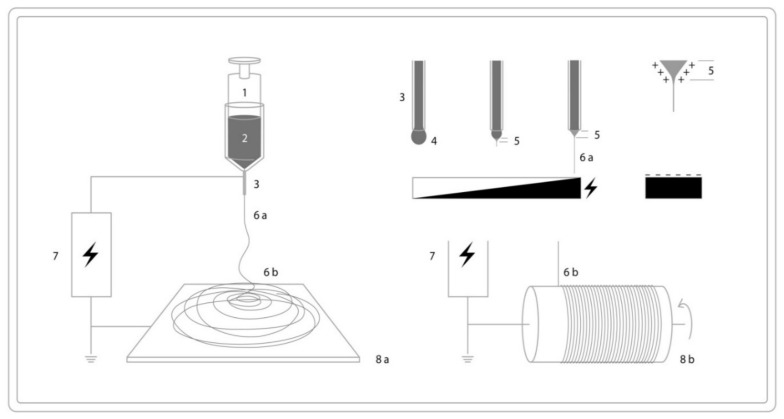
Schematic illustration of electrospinning setup: syringe (pump) (1), polymer solution (2), needle (3), liquid jet (4 and 6a), Taylor cone (5), fibers (6b), circuit (7), collector (8a) and drum collector (8b).

**Figure 6 polymers-13-01121-f006:**
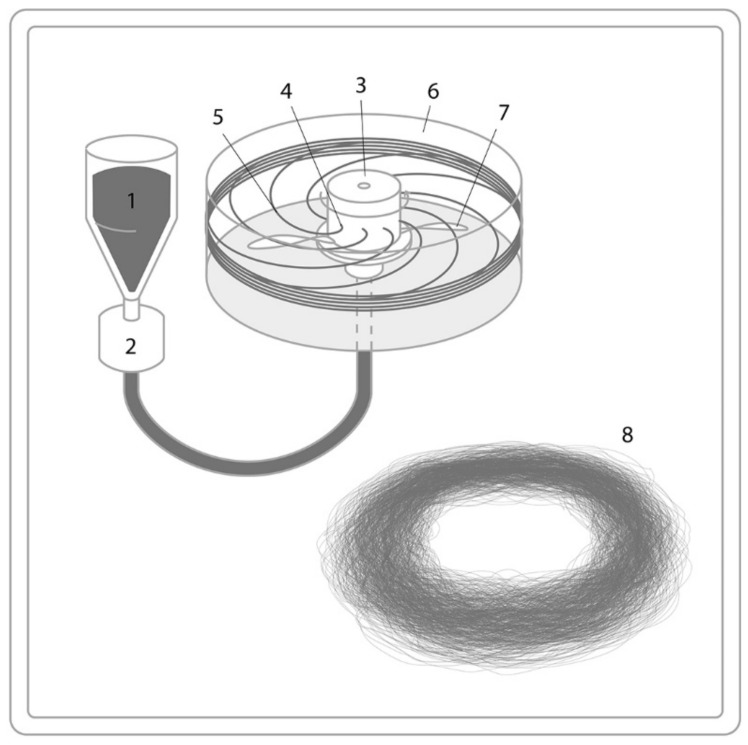
Centrifugal/rotary jet spinning setup with polymer (1), pump (2), polymer path (3), flexible air foil (4), rotating reservoir (6,7), and fibers (8).

**Figure 7 polymers-13-01121-f007:**
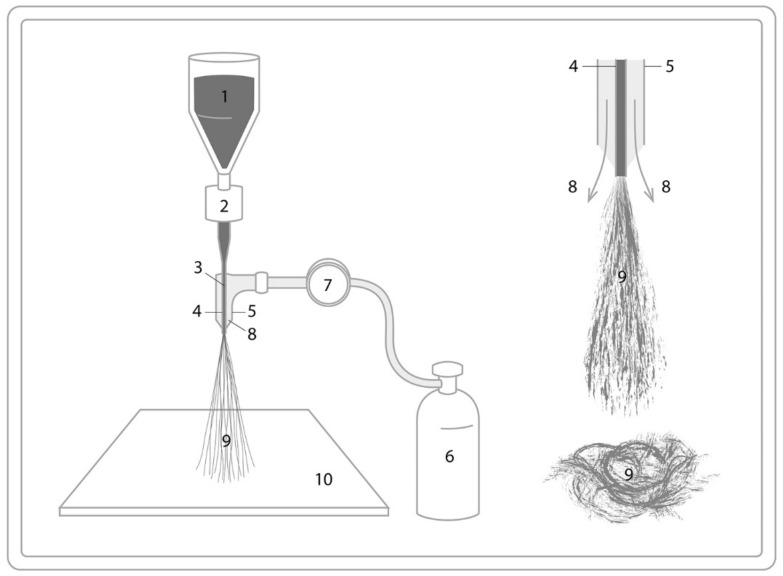
Schematic of the solution blowing process with the polymer (1), coaxial pump (2), syringe (3), polymer solution (4), air supply (5), high pressure air pump (5), valve (7), air (8), stretched polymer jet (9), and the fiber collector (10).

**Table 1 polymers-13-01121-t001:** Fiber spinning from thermoplastic starch composition (TPSC) [18].

Title	Process Used	Patent and Year
Method of making amylostic filaments and fibers	Mixing and extrusion of amylostic solids, water and glycerine at elevated temperature	US3499074 (1970)
Starch-containing fibers, process for their production	Mixing and extrusion of composition	US5516815 (1996)
Biodegradable fiber and non-woven fabric	Forming the fibers and bonding by heating at moist condition	US6045908 (2000)
Biodegradable fibers manufactured from thermoplastic starch.	Extrusion of the composition above into filaments, drawing, winding, and knitting	US6218321 (2001)
High elongation multicomponent fibers comprising starch	Extrusion of substituted starch and polymer having sheath–core configuration	US6623854 (2003)
Melt processable starch compositions	Mixing and extrusion of the composition	US6709526 (2004)
High elongation splittable multicomponent fibers comprising starch	Mixing extrusion and highspeed spinning of composition	US6743506 (2004)
Multicomponent fibers comprising starch and polymers	Melt spinning of the composition	US6746766 (2004)
Bicomponent fibers comprising a thermoplastic polymer surrounding a starch-rich core	Melt extrusion of composition and fiber formation	US6783854 (2004a)
Compositions and processes for reducing water solubility of a starch component in a multicomponent fiber	Melt spinning of the composition	US6830810 (2004b)
Fibers comprising starch and biodegradable polymers	Compounding and extrusion of composition and then melt spinning	US6890872 (2005)
Electro-spinning process for making starch filaments for flexible structure	Electrospinning of a starch/polymer (PAM) composition	US7029620 (2006)
Rotary spinning processes for forming hydroxyl polymer-containing fibers	Rotary spinning method forstarch/PVA composition	US7655175 (2010)
Starch fiber	Melt blowing of the composition	US7704328 (2010b)
Multicomponent fibers comprising starch and polymers	-----------------	US7851391 (2010)
Fibers comprising starch and a crosslinking agent	Melt blowing of the composition	US7938908 (2011)

**Table 2 polymers-13-01121-t002:** Fibers spun from non-thermoplastic starch composition (NTPSC) [18].

Title	Process Used	Patent and Year
Water-insensitive starch fibers and a process for the production thereof	Wet spinning of native and modified starch into an ammonium sulfate bath	US4139699 (1979)
Process for spinning starch fibers	Wet spinning of native starch and ammonium sulfate aqueous dispersion into ammonium sulfate bath	US4853168 (1989)
Fiber from blend of cellulose acetate and starch acetate	---------------------	US5446140 (1995)
Non-thermoplastic starch fibers and starch composition for making the same	-------------------	US6723160 (2004a)
Non-thermoplastic starch fibers and starch composition for making the same	---------------------	US6802895 (2004b)
Process for making non-thermoplastic starch fibers	Dry spinning of starch aqueous dispersion with high temperature attenuatingair flow	US6811740 (2004)
Non-thermoplastic starch fibers and starch composition for making the same	---------------------	US7025821 (2006)
Process for making non-thermoplastic starch fibers	Dry spinning of starch aqueous dispersion with high temperature attenuating air flow	US7276201 (2007)
Method for making polymeric structures	Making fiber products from starch, PVOH, and a crosslinking agent (imidazolidinone) and further adding of cellulose fibers	US7744791 (2010)
Method for forming fibers	Mixing, extrusion, and drawing of fibers from starch/PVA composition	US7939010 (2011)

**Table 3 polymers-13-01121-t003:** Electrospun fibers from starch [50].

Electrospun Material	Solvent Used	Characteristics of Obtained Fibers	Year of Publication
High amylose pure starch	Dimethyl sulfoxide DMSO/water	Fibers with diameter in range of micrometer	2014
Pure maize starch with (70%) amylose content	17% w aqueous formic acid solution	Diameter ranging from 80 to 300 nm	2013
High amylopectin corn starch and potato starch	2% w/w caustic soda solution	Submicron average diameter	2015
Waxy rice starch	water	Multiple flaky layers, highly porous	2016
High amylose modified starch 70% acetic anhydride	Ionic liquid 1-ally-1–3 methylimidazolium chloride	Continuous smooth fibers Diameter from 10 to 100 nm	2007
Modified maize starch with high amylose and 70% acetic anhydride	Formic acid	Tensile strength depends on starch to acetate ratio, annealing time, and degree of substitution	2009
High amylose modified maize starch 50% acetic anhydride	Dimethyl sulfoxide DMSO	Ultrafine fibers	2013
High amylose modified starch 70% formic acid solution	17% aqueous formic acid solution	Diameter ranging from 80 to 300 nm Elongation at break higher than native starch	2015
Acidified oxidized potato starch	DMSO	Smooth fibers at concentration up to 19%	2012
Natural tapiaco starch	Deionized water	Diameter from 1.3 to 14.5 µm	
Native and anionic corn starch	Formic acid	Fibers with diameter 70–264 nm	2018

**Table 4 polymers-13-01121-t004:** Electrospun fibers from starch/polymer blends [14,60].

Electrospun Materials	Solvent Used	Characteristics of Fibers Obtained	Year of Publication
Starch/PCL 30/70 wt.%	Acetic acid or chloroform	Diameter 130–180 nm Highly porous	2005
Starch/PCL 17% w/v	Chloroform/dimethyl formamide (DMF) (7:3)	Diameter from 400 nm to 1.4 µm	2010
Starch/PCL 30/70 wt.%	Chloroform/DMF (7:3)	Diameter approximately 400 nm fine morphology	2008
Starch/PCL 30/70 wt.%	Chloroform 40% w/v	Fiber diameter around 100 µm	2010
Potato starch (5 wt.%)/polyvinyl alcohol/PVA	Ethanol 5 wt.%		2010
Soluble starch/PVA 1:1 or 1:3	Water	Good morphology	2014
Oxidized starch (OS)/PVA	Water	Diameter affected by weight ratio of PAV/OS	2011
Cationic starch (CS)/PVA (3:1)	Water		2012
Cationic starch (CS)/PVA	Ethanol/water	Thicker and stick nanofiber	2009
Starch/poly (lactide–co-glycolide) (PLGA)	Starch in DMSO and PLAGA in tetrahydrofuran (THF)/N		2011
Cassava starch/PLA	PLA in dichloromethane, cassava starch in DMSO	Smooth fibers	2011
High amylose maize starch, cationic starch and Nanocellulose	Dimethyl sulfoxide and ethanol	Good strength	2018
Rice starch/PVA (25 wt.%)	Water and NaoH	Uniform fibers with diameter 36–151 nm	2017
Glutinous rice starch/PVA (2 w/v and 8 w/v)	Hot water	Smooth morphology with diameter 191–263 nm	2017
Starch formate/glycerol (17 wt.%)	Formic acid	Fibers with diameter 4.13 µm	2017
Corn starch/guar gum (3 wt.%)	Water	Fibers with diameter 95 nm	2017
Starch acetate (20 wt.%)	Formic acid/water (90:10 v/v)	Fibers with good tenacity and uniformity	2009
Carboxymethyl starch/PLA	Sodium dodecyl sulfate	Diameter 190–265 nm	2019

**Table 5 polymers-13-01121-t005:** Comparison of the different fiber spinning technologies [64,65,66,67].

Fiber Spinning Methods	Advantages	Disadvantages
Melt spinning	simple and versatile process; as lower manufacturing cost; ability to control fiber diameter; high output compared to solution spinning; no solvent needed.	risk of thermal degradation due to high shear rates; variable diameters distribution; high energy consumption; limited to large fiber diameter.
Solution Spinning (dry and wet)	low energy consumption; variety of structures; pre-controlled fiber diameter.	removal of solvent required; complex coagulation baths; difficult to obtain aligned fibers; environmental impact of solvents.
Melt electro-spinning	solvent free process; low process costs; long and continuous fibers.	low outputs; limited varieties of polymer; risk of blocking the needle; risk of thermal degradation; electric discharge problems.
Solution electrospinning	used for long and continuous fibers; easy to set up; uniform in diameter; diversified in composition.	requires solvent recovery; risk of toxic contamination; jet instability; low output for single spinneret.
Centrifugal spinning	simple, safe, and eco-friendly process; free from high electric field and solvents; high production rates; low costs.	might require high temperature.
Solution blow spinning	large-scale fabrication; wide range of materials; good industrialization prospect.	random deposition; poor fiber morphology.
Template synthesis	simple and one step process; suitable for longer fibers; easy to change diameter by using different templates.	lower productivity; non-uniform fiber size/shape; complex process.
Self-assembly	easy to obtain smaller nanofibers.	complex process.
Melt blowing	long and continuous fibers; high productivity; no need of solvent recovery.	polymer limitations; risk of thermal degradation; expensive process; complex setup needed
Phase separation	simple equipment required.	only works with selective polymers.

## Data Availability

The data presented in this study are available in the below listed references.

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
