# Peer review of "Review on Spinning of Biopolymer Fibers from Starch"

_polymers, 2021, doi:10.3390/polym13071121_

Round 1

Reviewer 1 Report

The authors reviewed the spinning techniques for bio-based polymers particularly for starch and its blend compositions and provide product examples. I found the review interesting, timely, and adequate. The review is worth for publication. However, structurally I found that there is a misalignment. I would suggest the authors to re-structure introduction and conclusion sections since the conclusion provides introductory information more than providing a summary of what they have presented in the review. More clear messages related to the results they have discussed in literature would be helpful for the readers. Also, please provide a reference including a title for the ongoing work that is mentioned in the conclusion. Moreover, I believe that the paper recently published, Sustainability 202113(3), 1160; https://doi.org/10.3390/su13031160,  would fit nicely to this review especially at the end of first paragraph of introduction.

Reviewer 2 Report

I have reviewed a manuscript entitled “REVIEW ON SPINNING OF BIO-POLYMER FIBERS FROM STARCH”. It is a well-written and interesting study that can attract the reader’s attention. I believe it is suitable for the publication after considering the following comments:

Comment 1: Please add more about the application of bio-based materials in replacing the oil-based materials in different applications. Following references might be helpful:

https://www.sciencedirect.com/science/article/pii/S0300944019304138

https://www.sciencedirect.com/science/article/pii/S0300944019304126

https://www.mdpi.com/2073-4360/12/6/1234

https://onlinelibrary.wiley.com/doi/full/10.1002/anie.201410770

https://www.mdpi.com/2073-4360/8/7/262

Comment 2: Could you please add a section and discuss the usage of starch biopolymers in various technological applications.

Comment 3: Could you please compare the advantages and disadvantages of the different methods of spinning in a table?

Comment 4: I would suggest adding a catchy and appealing figure instead of figure 1, showing the importance of the starch biopolymers.  

Reviewer 3 Report

The effects of oil-based plastics and additives on climate, water and soil quality are omnipresent. In the last few decades, many scientific studies and developments have been undertaken in order to manufacture products sustainably from renewable raw materials and thus reduce the dependence on fossil raw materials and the global amount of microplastics. Textiles, based on synthetic fibers are supposed to be one of the biggest sources for microplastic. In addition, increasing interest in bio-based polymers and fibers has led to the development of several alternatives to conventional plastics and fibers. Biopolymer fiber can be made from renewable, environmentally friendly resources and be fully biodegradable. Biogenic resources with high content of carbohydrates like starch-containing plants have huge potentials to substitute conventional synthetic plastics in a number of applications. In this paper, possible spinning techniques employed for the development of virgin starch or starch/polymer blend fibers and their products are discussed. Beneficiation of starch for the development of biobased fibers can result in sustainable replacement of oil-based high-value materials with cost-effective, environmentally friendly and abundant products. The topic is important, the results are interesting and the methodology followed is appropriate, while the content falls well within the scope of this Journal. Besides, the layout is clear and easy to understand. In general the paper makes fair impression and my recommendation is that it merits publication in this Journal, after the following major revision:

  1. The detailed literature review indicates efforts made by the authors. The coherence of the related work, however, is still not clear. It may help the authors by answering the following questions: Why are these works relevant? Which specific problems were addressed? How are the previous results related with the latest work? What are the outstanding, unresolved, research issues? Which of them has been solved by the proposed study? Answering the questions leads to the novelty of the proposed work naturally. Besides, the current one is nothing but a literature review. Why their work is important comparing to previous reports? I think this is essential to keep the interest of the reader.
  2. Extensive research and development work are required to develop appropriate spinning technologies for their full utilisation as a source of biobased high value-product. The authors should give some explanation on above conclusions. What the authors meant is not clear to reader.
  3. There is no sufficient literature on characterization of starch-based fibers produced using melt spinning. Really?
  4. Please expand the motivation, the problem context, clarify the problem description, and (if possible) add specific objectives.
  5. Polymer fiber including fibrous porous media has been widely used in many fields of life. The present review mainly focuses on spinning techniques. It does not necessarily imply that the theoretic work is not important. The authors omit this part during the current literature review, which should include a brief review of the theoretic work after revision. In the theoretic perspective, fractal models are frequently used for characterization of properties of polymer fiber including fibrous porous media (see [A fractal model for capillary flow through a single tortuous capillary with roughened surfaces in fibrous porous media, Fractals, 2021, 29(1):2150017; Fractals, 2019, 27(7): 1950116; Powder Technology, 2019, 349:92-98]). Authors should introduce some related knowledge to readers.
  6. Please, expand the conclusions in relation to the specific goals and the future work.

Round 2

Reviewer 2 Report

I think it is suitable for publication in the present form. 

Reviewer 3 Report

My comments have been addressed.